**Investigation**

# Allele ages provide limited information about the strength of negative selection

Vivaswat Shastry [ID],[1] Jeremy J. Berg[2],*

[1]Committee on Genetics, Genomics and Systems Biology, University of Chicago, Chicago, IL 60637, USA
[2]Department of Human Genetics, University of Chicago, Chicago, IL 60637, USA

*Corresponding author: Department of Human Genetics, University of Chicago, Chicago, IL 60637, USA. Email: jjberg@uchicago.edu

For many problems in population genetics, it is useful to characterize the distribution of fitness effects (DFE) of de novo mutations among a certain class of sites. A DFE is typically estimated by fitting an observed site frequency spectrum (SFS) to an expected SFS given a hypothesized distribution of selection coefficients and demographic history. The development of tools to infer gene trees from haplotype alignments, along with ancient DNA resources, provides us with additional information about the frequency trajectories of segregating mutations. Here, we ask how useful this additional information is for learning about the DFE, using the joint distribution on allele frequency and age to summarize information about the trajectory. To this end, we introduce an accurate and efficient numerical method for computing the density on the age of a segregating variant found at a given sample frequency, given the strength of selection and an arbitrarily complex population size history. We then use this framework to show that the unconditional age distribution of negatively selected alleles is very closely approximated by reweighting the neutral age distribution in terms of the negatively selected SFS, suggesting that allele ages provide little information about the DFE beyond that already contained in the present day frequency. To confirm this prediction, we extended the standard Poisson random field method to incorporate the joint distribution of frequency and age in estimating selection coefficients, and test its performance using simulations. We find that when the full SFS is observed and the true allele ages are known, including ages in the estimation provides only small increases in the accuracy of estimated selection coefficients. However, if only sites with frequencies above a certain threshold are observed, then the true ages can provide substantial information about the selection coefficients, especially when the selection coefficient is large. When ages are estimated from haplotype data using state-of-the-art tools, uncertainty about the age abrogates most of the additional information in the fully observed SFS case, while the neutral prior assumed in these tools when estimating ages induces a downward bias in the case of the thresholded SFS.

**Keywords:** frequency spectrum; DFE; genealogy; MAF; ARG

## Introduction

Understanding how natural selection has shaped patterns of genetic variation is a core goal in population genetics (Fisher 1930; Wright 1931; Haldane 1932). One valuable way of developing this understanding is by learning about a distribution of fitness effects (DFE), i.e. the probability distribution on the selection coefficients of newly arising mutations. (Eyre-Walker and Keightley 2007; Boyko et al. 2008). There are several approaches to estimating the DFE. For species amenable to laboratory experimentation, the DFE can be estimated directly by comparing the fitness of individuals carrying different mutation complements (Fowler et al. 2010; Hietpas et al. 2011; Bataillon and Bailey 2014; Boucher et al. 2014), or indirectly from changes in the fitness of lines of laboratory organisms during mutation accumulation experiments (Eyre-Walker and Keightley 2007; Halligan and Keightley 2009). In many organisms (e.g. humans), this is not possible, and the DFE is estimated from polymorphism data using population genetic models (Bustamante et al. 2001; Williamson et al. 2005; Kim et al. 2017; Johri et al. 2022).

These approaches typically use the site frequency spectrum (SFS)—a vector recording the number of mutations observed at

each possible sample frequency—as a summary statistic for estimating the DFE under the Poisson random field (PRF) model first introduced by Sawyer and Hartl (1992), and elaborated by several subsequent authors (see, e.g. Keightley and Eyre-Walker 2007; Tataru et al. 2017; Sendrowski and Bataillon 2024). Intuitively, because negative selection against a mutation reduces the probability that it will climb to a high frequency in the population, and thus be observed at a high frequency in the sample, information about the DFE comes from the shape of the SFS. However, the frequency of a single variant provides relatively little information about its selection coefficient, requiring that information be pooled across a relatively large number of sites to estimate a DFE (Williamson et al. 2005; Boyko et al. 2008; Kim et al. 2017). As a consequence, DFE inferences typically focus on relatively large classes of sites (e.g. all nonsynonymous variants), and efforts to estimate the DFE for smaller subsets of sites (e.g. variants associated with a particular complex trait, (Simons et al. 2022) or the set of loss-of-function mutations in a gene, (Zeng et al. 2023)) come with significant uncertainty.

One plausible way of overcoming this limitation is to include more information about the past frequency trajectory of each variant into the inference. To this point, breakthroughs in

coalescent inference methods over the past 10 years have enabled the estimation of gene trees from DNA sequence data (Rasmussen *et al.* 2014; Kelleher *et al.* 2019; Speidel *et al.* 2019; Albers and McVean 2020; Lewanski *et al.* 2024), while the increase in the amount of ancient DNA sequenced over roughly the same time period has allowed for temporal sampling of allele frequencies. Recent work has leveraged both estimated genealogies and ancient DNA time series to estimate positive selection coefficients for individual beneficial mutations (Malaspinas *et al.* 2012; Mathieson and McVean 2013; Stern *et al.* 2019; Irving-Pease *et al.* 2024; Mathieson and Terhorst 2022; Vaughn and Nielsen 2024).

By contrast, methods for incorporating such information into the DFE inference paradigm for sites under negative selection have largely done so implicitly, using custom-built models and/or simulations to link patterns of haplotypic variation to the DFE for deleterious mutations (see e.g. Johri *et al.* 2020; Ortega-Del Vecchyo *et al.* 2022). Outside of the traditional DFE inference paradigm, one simple summary of an allele's past trajectory, its estimated age, has been used explicitly to study the impact of negative selection on complex trait associated loci from genome-wide association studies (GWAS). In this context, allele age estimates have been used to develop annotations that explain spatial variation across the genome in the magnitude of the contribution to complex trait variance (Gazal *et al.* 2017, 2018; Hujoel *et al.* 2019; Kichaev *et al.* 2019; Kanai *et al.* 2021; Shi *et al.* 2021; Nait Saada *et al.* 2023). The sign of this association is consistently negative across traits, such that younger alleles contribute more to heritability, consistent with the impact of negative selection. Notably, this association is not explained by the mutual association of heritability and age with minor allele frequency, suggesting that the ages may contain substantial additional information about the impact of selection beyond that which is included in the allele frequency. However, we still lack a complete understanding of the relationship between frequency, age, and strength of selection, limiting our ability to clearly interpret these associations.

In this paper, we study the utility of including allele age into the inference of deleterious fitness effects. Allele ages represent an attractive choice for this purpose because (1) when combined with the present-day frequency, they would seem to summarize much additional information about an allele's past trajectory in a single data point, and (2) there has already been much theoretical and empirical work on the distribution of allele ages, which we can build on. For example, derived alleles under selection (whether positive or negative) are known to be younger on average than neutral alleles at the same frequency in the population (Kimura and Ohta 1973; Maruyama 1974; Kiezun *et al.* 2013, also see Supplementary Fig. S1). As famously shown by Maruyama (1974), after conditioning on the present-day frequency of the allele in the population, the distribution of ages for alleles under positive and negative selection of the same magnitude are identical, though this symmetry breaks down if we condition instead on sample frequency (Stephens and Donnelly 2003), or if the population size is not constant over time (Ortega-Del Vecchyo *et al.* 2022).

Several authors have developed tools to approximate or simulate from the distribution of allele ages, either under neutrality (Rannala 1997; Griffiths and Tavaré 1998; Slatkin 2000; Slatkin and Rannala 2000), or in the presence of selection (Slatkin 2001; Wiuf 2001; Stephens and Donnelly 2003). A general expression for the density on the age of selected alleles is given by Griffiths (2003) in terms of the fixation probability and transition density of the Wright–Fisher diffusion, though evaluating this expression requires an appropriate numerical approximation of the transition density (e.g. Song and Steinrücken 2012). If the population

size is constant through time, samples from the distribution of allele age conditional on the population frequency can be obtained by simulating allele frequency trajectories forward in time, starting from the desired frequency and recording the time until loss. Then, due to the time reversibility of the Wright–Fisher diffusion, the distribution of these times until loss represent valid samples from the distribution of allele ages (Slatkin 2001). When the population size is not constant, this time reversibility no longer holds and this method is not valid. In this case, samples from the age distribution can still be obtained by simulation, either via a brute force forward-in-time approach (using a tool like PReFerSim, Ortega-Del Vecchyo *et al.* 2016), or via an importance sampling approach introduced by Slatkin (2001). However, all of these simulation based approaches entail substantial computational cost and/or the potential for error due to the Monte Carlo approximation.

Here, we first build on the moments framework developed by Jouganous *et al.* (2017) to introduce an accurate and efficient numerical method for computing the distribution on the age of a segregating variant given its sample frequency, selection coefficient, and an arbitrarily complex population size history. Second, we use this framework to show that the age distribution of negatively selected alleles with a given scaled selection coefficient can be closely approximated by reweighting the age distribution of neutral variants across allele frequency bins by the ratio of the normalized SFS entries for the deleterious and neutral variants. Notably, the same is not true for positively selected variants, where information about the past frequency trajectory of an allele has proven extremely valuable (Hejase *et al.* 2020). This observation suggests that, if the full distribution of allele frequencies is observed, then allele ages should carry relatively little information about negative selection coefficients beyond that which is already contained in allele frequency data, in contrast to the case for positively selected variants. To verify this prediction, we extend the standard PRF model for estimating fitness effects from distributions of allele frequencies to include the joint distribution of frequency and age, using our numerical method. Finally, we use additional simulations to show how allele ages can provide useful information about negative selection coefficients if only alleles above a certain minor allele frequency cutoff are observed (e.g. as in GWAS) or if the sample is small.

## Results

### A numerical method for the density on allele ages

First, we develop a numerical method to compute the distribution on allele age for a segregating variant under selection, conditional on its sample frequency. To do this, we build on a numerical approximation to the Wright–Fisher diffusion developed by Jouganous *et al.* (2017) (see also Evans *et al.* 2007 and Malaspinas *et al.* 2012, for closely related earlier work), which allows us to compute this density on age using an efficient dynamic programming algorithm.

We consider a large population of $N$ diploid individuals under a model consistent with the Wright–Fisher diffusion with selection. We imagine tracking the evolution of a very large number of independent sites, each with a small mutation rate, such that an infinite sites approximation applies. At each site, we follow a sample of $2n$ lineages through time (assuming $n \ll N$), and track the number of lineages at each site that carry a derived allele. Using the moment recursions developed by Jouganous *et al.* (2017), we can track the expected number of sites where the derived allele will be found $i$ times in the sample, i.e. the expected SFS. We write $\Phi_{2n}^t$ to denote the length $2n - 1$ vector which records this expected

SFS in generation $t$, where $t$ counts down as time moves forward until $t = 0$ at the present generation. The ith entry of $\Phi_{2n}^t$ therefore gives the expected number of sites with a derived allele at frequency $i$ in the sample of size $2n$, $t$ generations before the present. Given $\Phi_{2n}^t$, we can obtain the expected SFS in generation $t - 1$ via two steps. First, common ancestor events (i.e. genetic drift/coalescence) and selection events lead to the movement of mass among adjacent bins of the SFS (along with loss of mass from the singleton and "$2n - 1$"-ton bins) from one generation to the next. Second, new mutations are expected to arise at frequency $i = 1$ at $2n\mu_t$ sites that previously lacked segregating derived alleles, where $\mu_t$ is the total mutation rate across all sites in generation $t$. These dynamics are described by

$$\Phi_{2n}^{t-1} = (I + \Xi_{2n,t})\Phi_{2n}^t + 2n\mu_t\delta_1 \tag{1}$$

where $I$ is the identity matrix, $\Xi_{2n,t}$ is a tri-diagonal matrix containing the coefficients describing how the combination of genetic drift and selection operating in generation $t$ move mass between adjacent bins in the expected SFS, and $\delta_1$ denotes a vector with a 1 in the singleton bin and 0 elsewhere. The coefficients of $\Xi_{2n,t}$ are precisely as given by Jouganous et al. (2017), including their jackknife approximation to close the moment equations in the presence of selection (we reproduce these coefficients in Supplementary Section S2). Notably, differences in the population size or strength of selection across generations are accounted for via differences in the coefficients of $\Xi_{2n,t}$.

The probability distribution on the allele age, $a$, given an arbitrary sequence of selection coefficients, population sizes, and mutation rates, can be written as

$$P\left(a \mid i, n, \{s_0, \ldots, s_{T_{max}}\}, \{N_0, \ldots, N_{T_{max}}\}, \{\mu_0, \ldots, \mu_{T_{max}}\}\right) = \frac{m_a[i]}{\Phi_{2n}^0[i]}, \tag{2}$$

where $m_a[i]$ is the expected number of mutations that arise in generation $a$ and are found at frequency $i$ in the present day. Notably, $\Phi_{2n}^0[i] = \sum_{a=1}^{T_{max}} m_a[i]$, i.e. the ith entry in the present-day frequency spectrum is simply a sum over contributions from mutations that arose in all prior generations. We first compute the denominator, $\Phi_{2n}^0[i]$, by initializing $\Phi_{2n}^{T_{max}}[i] = 0$ for all $i$, with $T_{max}$ set sufficiently far in the past that a negligible number of mutations arising before this time would be expected survive to the present day. We then iterate Equation (1) until we obtain $\Phi_{2n}^0$. To compute the numerator, we imagine tracking the sample frequency of the derived allele at each of the $\Phi_{2n}^0[i]$ sites found at frequency $i$ in the present, backward in time until for each one we encounter the mutation event from which it arose. Concretely, we write $\Psi_{i,2n}^t$ for the expected SFS of this conditional sample at time $t$, initializing

$$\Psi_{i,2n}^0[j] = \begin{cases} \Phi_{2n}^0[j], & \text{if } j = i \\ 0, & \text{otherwise,} \end{cases} \tag{3}$$

at generation 0. We then evolve this conditional SFS backward in time by iterating

$$\Psi_{i,2n}^t = (I + \Xi_{2n,t})^{-1}\left(\Psi_{i,2n}^{t-1} - \delta_1 m_{t-1}[i]\right) \tag{4}$$

from $t = 1$ to $t = T_{max}$, where

$$m_t[i] = 2n\mu_t\frac{\Psi_{i,2n}^t[1]}{\Phi_{2n}^t[1]} \tag{5}$$

(see Supplementary Section S3). The ratio $\Psi_{i,2n}^t[1]/\Phi_{2n}^t[1]$ gives the expected fraction of mutations found as singletons in a sample taken in generation $t$, which are destined to be found at frequency $i$ in the present, and can thus be interpreted as the probability that a random singleton at generation $t$ will be found at frequency $i$ in the present. The product of this probability with the number of mutations arising in generation $t$ therefore gives expected number of mutations that arise in generation $t$ and are found in the sample at frequency $i$ in the present.

Notably, if the population size, selection coefficient, and mutation rate are constant over time, then $\Phi_{2n}^t[1]$ attains a steady-state value, $\Phi_{2n}[1]$. In this case, Equation (2) reduces to

$$P\left(a \mid i, n, \gamma\right) = \frac{\Psi_{i,2n}^a[1]}{\sum_{t=1}^{T_{max}} \Psi_{i,2n}^t[1]}, \tag{6}$$

(where $\gamma = 2Ns$ is the population scaled selection coefficient). This follows from the reversibility of the Wright–Fisher diffusion with respect to its steady state, and is related to the observation that the age distribution in a constant size population (with a constant selection coefficient and mutation rate) can be obtained by simulating forward in time until loss and then reversing the trajectory (Maruyama 1974; Slatkin 2001), while this is not possible in a population that varies in size, or if the selection coefficient or mutation rate vary over time. In principle, this suggests that if all three of these parameters are held constant, then the initial forward pass through time can be skipped, and that the age distribution can be obtained with only a single backward pass through time.

However, computing the $\Psi_{i,2n}^t[1]$ for each generation still requires that we compute the $m_{t-1}[i]$, so we still need to know the value of $\Phi_{2n}^t[1]$. In principle, for constant size population under additive selection, $\Phi_{2n}^t[1]$ could be obtained by integrating the analytical expression for the population SFS (Wright 1938; Bustamante et al. 2001) against the binomial probability of obtaining a singleton in the sample, but here we simply run the algorithm forward in time as in the nonequilibrium case, given that doing so is not computationally expensive.

This method provides a means to compute the density on allele age that is both accurate and efficient. For example, computing the distribution of ages for a single $i$ in a sample of $n = 125$ and selection coefficient $s = -5 \times 10^{-4}$ assuming the piecewise constant model of exponential growth inferred by Tennessen et al. (2012) for African–American individuals (over 55,000 generations, shown in Supplementary Fig. S2) takes on average 4 s on a MacBook Pro M1 (2021). In Supplementary Fig. S3, we validated the accuracy of the method by comparing the cumulative distribution of ages conditional on segregation in a sample of $n = 125$ to that obtained via forward-in-time simulations using PReFeRSim (Ortega-Del Vecchyo et al. 2016) under two scenarios: neutrality ($s = 0$) in a population of constant size, and moderate negative selection ($s = -5 \times 10^{-4}$) in the Tennessen et al. (2012) growth model. In both cases, the distribution obtained with our method closely matches the simulations (see Supplementary Fig. S3). Notably, although the method is fast to obtain the age distribution for a single sample frequency, obtaining age distributions for all sample frequencies, takes $2n$ times as long, because the age distribution must be computed separately for each sample frequency. This can become substantial, especially if sample sizes are large (e.g. $\sim 1,000$ s for the example case with $n = 125$ considered here). In Supplementary Section S1, we outline an alternative algorithm to obtain the age distribution for all sample frequencies at the

same time in a more efficient manner. Briefly, this method relies on the fact that the probability that an allele found at frequency $i$ is $a$ generations old is proportional to the number of such mutations present in the sample. Thus, we can efficiently compute the density on allele age across all frequencies by first computing the sample SFS conditional on allele age, forward in time, for each generation in which a mutation could have arisen. For the example case considered here, this method takes approximately 30 s to obtain all $2n - 1 = 249$ age distributions. Both methods scale linearly in sample size and $T_{max}$, and the runtimes for different combinations of parameters are shown in Supplementary Fig. S4.

## The age distribution of selected alleles

We next applied our method to study how selection impacts the distribution on ages, conditional on segregation in a present-day sample. In this section, we focus largely on the constant size case, so we represent time in coalescent units and the strength of selection in terms of the population scaled selection coefficient, $\gamma = 2Ns$, and we write the density on age given a particular scaled selection coefficient as $P(a \mid i, n, \gamma)$. In Fig. 1, we plot age distributions in a diploid sample of size $n = 125$, across a range of selection coefficients and sample frequencies. As expected, for a given sample frequency, alleles under stronger selection are younger on average with less variation in age than less strongly selected alleles (Maruyama 1974; Wiuf 2001; Griffiths 2003; Stephens and Donnelly 2003, also see Supplementary Fig. S1). More precisely, for a given sample frequency, the allele age distribution for sites under weaker selection is generally very similar to the age

distribution for neutral alleles with the same sample frequency, with significant deviations arising only when selection is stronger.

This pattern is closely related to the impact of selection on the SFS. To this point, the probability that the frequency of a derived allele with scaled coefficient $\gamma$ is found in the $[x, x + dx]$ frequency interval in the *population* is proportional to

$$P(x \mid \gamma) \propto \frac{e^{-2\gamma(1-x)}}{x(1-x)}$$

(Wright 1938). To a first approximation, this density on population allele frequencies is proportional to $1/x$, i.e. that of neutral alleles, at frequencies below a threshold value of $x_\gamma^\star = 1/2\gamma$, but drops quickly to zero for frequencies above this threshold when selection is negative ($\gamma < 0$; see Supplementary Fig. S5). If the sample size is large relative to this threshold on the population frequency (i.e. if $2nx_\gamma^\star \gg 1$), then the sample SFS is expected to show a similar pattern, resembling that of neutral alleles for allele counts satisfying $i/2n \ll x_\gamma^\star$, while sites with allele counts satisfying $i/2n \gg x_\gamma^\star$ are rarely observed. This reasoning suggests as long as the sample is large, we should expect $x_\gamma^\star$ to mark the approximate boundary between sample frequencies for which the age distribution is similar to that of neutral alleles, and those for which it is not (this will not hold in small samples, which we consider separately later on).

To test this prediction quantitatively, we compute the Kullback–Leibler (KL) divergence between the conditional age distribution of selected alleles at a given sample frequency and the corresponding age distribution of neutral alleles at the same sample frequency. In general, the KL divergence between some true

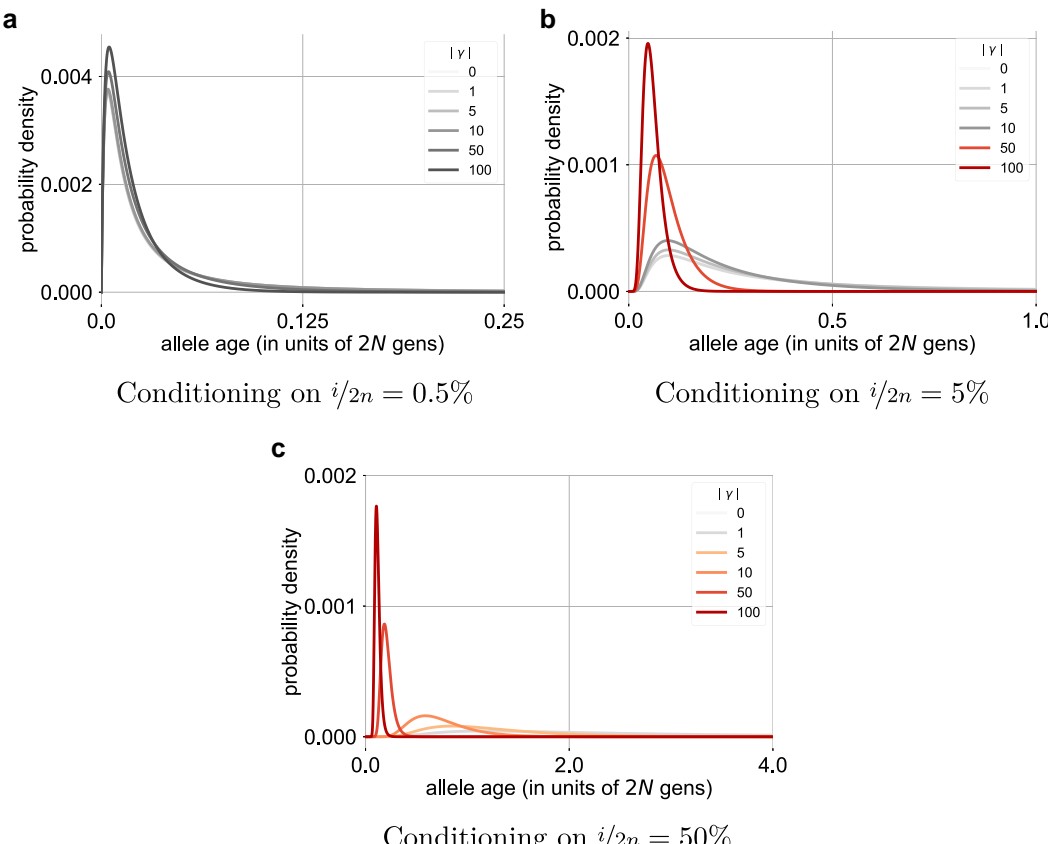

**Fig. 1.** Age distributions under different strengths of selection, conditional on segregating at a particular frequency in a diploid sample of $n = 125$. For a particular sample frequency $x^\star$, scaled selection coefficients less than $1/2x^\star$ are shown in gray-scale, while the selection coefficients that are larger than this threshold are shown in color. a) Conditioning on $i/2n = 0.5\%$, b) Conditioning on $i/2n = 5\%$, and c) Conditioning on $i/2n = 50\%$.

distribution $p$ and an approximation $q$ is the expectation over samples taken from the true distribution of the log-likelihood ratio of the true distribution to the approximation. That is, $D_{KL}(p \parallel q) = \mathbb{E}_p[\log \frac{p}{q}]$. Thus, exponentiating the KL divergence tells us, on a per site basis, how much more likely the data are on average under the generating distribution ($p$) than under the approximation ($q$). To facilitate this interpretation, we measure the KL divergence in base 10 so that a divergence of $k$ means that the data are $10^k$ times more likely under the generating distribution than the approximating distribution for alleles at the same frequency.

We compute the divergence between selected and neutral age distributions, conditional on sample frequency, as

$$D_{KL}\big(P(a \mid i, \gamma, n) \parallel P(a \mid i, \gamma = 0, n)\big)$$
$$= \sum_{a=1}^{T_{max}} P(a \mid i, \gamma, n) \log_{10}\left(\frac{P(a \mid i, \gamma, n)}{P(a \mid i, \gamma = 0, n)}\right). \quad (7)$$

In Fig. 2, we plot this divergence as a function of the scaled selection coefficient $\gamma$ and sample count $i$, assuming that $n = 100$. Confirming our prediction, we find that broadly, across the range of scaled selection coefficients, the distance is close to zero for sample frequencies less than $x^\star_\gamma = 1/2\gamma$ (shown by the dashed black line), but begins to increase above this value. We also compute the total variation distance between the two distributions (interpretable as being proportional to the $\ell_1$ norm and as a bounded measure between 0 and 1), and show the result in Supplementary Fig. S6.

This observation suggested that it should be possible to closely approximate the unconditional age distribution of negatively

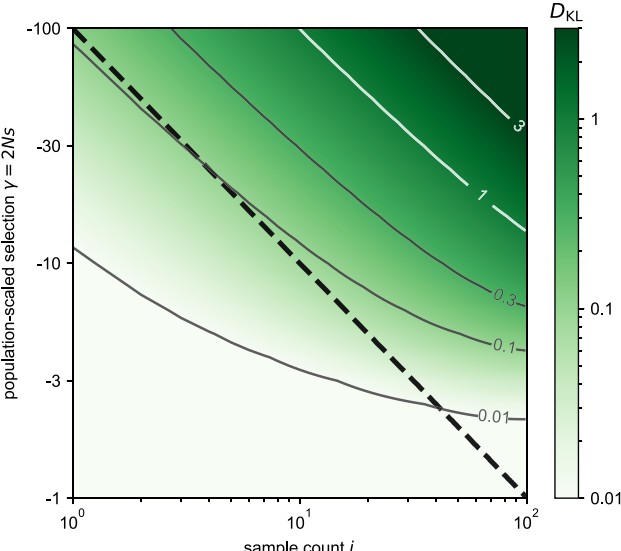

**Fig. 2.** The heatmap of KL divergence between the age density of alleles given a particular selection coefficient, $P(a \mid i, \gamma, n)$, and that of neutral alleles, $P(a \mid i, \gamma = 0, n)$, conditional on the sample allele count $i$ (Equation (7)). The black dashed line indicates the threshold, $\gamma^\star = 1/2x^\star_\gamma = n/i$, above which we expect the conditional age distribution of selected alleles to differ substantially from that of neutral alleles. Values in this figure are calculated for a diploid sample size of $n = 100$, so a sample count $i = 100$ corresponds to a sample frequency of 50%. Following the intuition from the text, an isocline of 0.3 reflects an approximately 2× increase in likelihood of the ages coming from a selected distribution than the neutral one. (See Supplementary Fig. S6 for the analog with total variation distance.)

selected alleles, $P(a \mid \gamma, n)$, by a simple reweighting of the neutral age distribution in terms of the SFS for negatively selected alleles. Intuitively, we imagine first sampling pairs of frequencies and ages for a large number of neutral alleles. We then imagine retaining each sampled site with probability proportional to

$$w_i = \frac{P(i \mid \gamma, n)}{P(i \mid \gamma = 0, n)}, \quad (8)$$

where $P(i \mid \gamma, n)$ and $P(i \mid \gamma = 0, n)$ are the sample SFSs, conditional on segregation, for selected and neutral alleles, respectively. The resulting distribution of allele frequencies among the resampled sites will then be a valid sample from the distribution of frequencies under selection (that is, from $P(i \mid \gamma, n)$). Within each frequency bin, the ages distribution still follows the neutral distribution, $P(a \mid i, \gamma = 0, n)$, but resampling the SFS to match the selected SFS largely suppresses the contributions from the frequency bins where the age distributions of neutral and selected alleles diverge. The approximate age distribution implied by this resampling procedure can be computed directly as

$$P(a \mid \gamma, n) \approx \tilde{P}(a \mid \gamma, n) = C^{-1} \sum_{i=1}^{2n-1} w_i P(a \mid i, \gamma = 0, n), \quad (9)$$

where $C = \sum_{i=1}^{2n-1} w_i$ is a renormalization constant which ensures that $\sum_{a=1}^{T_{max}} \tilde{P}(a \mid \gamma, n) = 1$. Supplementary Fig. S7 illustrates the resampling weights, $w_i$, for a few different choices of $\gamma$ around 0.

In Fig. 3, we compare this approximation to the exact unconditional age distribution, which we compute using our numerical framework as

$$P(a \mid \gamma, n) = \sum_{i=1}^{2n-1} P(a \mid i, \gamma, n). \quad (10)$$

Figure 3a shows this comparison for a representative case of moderately strong negative selection ($\gamma = -20$). In this case, the approximation in Equation (9) closely matches the truth in Equation (10), with significant differences emerging only for older alleles which are unlikely to be found under either the true model or the neutral resampling approximation (see Supplementary Fig. S8 for the constant population size case and Supplementary Fig. S10 for the human-like exponential growth over a range of selection strengths). This pattern indicates that the difference between the unconditional age distribution for neutral alleles and those under moderate negative selection is mostly explained by the relative dearth of high-frequency alleles under negative selection, and not by the differences in the conditional age distributions.

In contrast, for a complementary case of moderate positive selection ($\gamma = 20$; Fig. 3b), Equation (9) provides a poor approximation of the true unconditional age distribution. The reason is that positive selection amplifies the abundance of precisely those frequency bins for which the conditional age distributions are most strongly shifted relative to the neutral expectation. As a result, the neutral resampling distribution places greater mass on older ages compared with the true distribution, given the abundance of higher frequency alleles in this positively selected case compared with the previous negatively selected case.

To quantify the accuracy of this approximation, we again compute a KL divergence, this time between the approximation given in Equation (9) and the exact unconditional age distribution in Equation (10). The relevant divergence is then given by

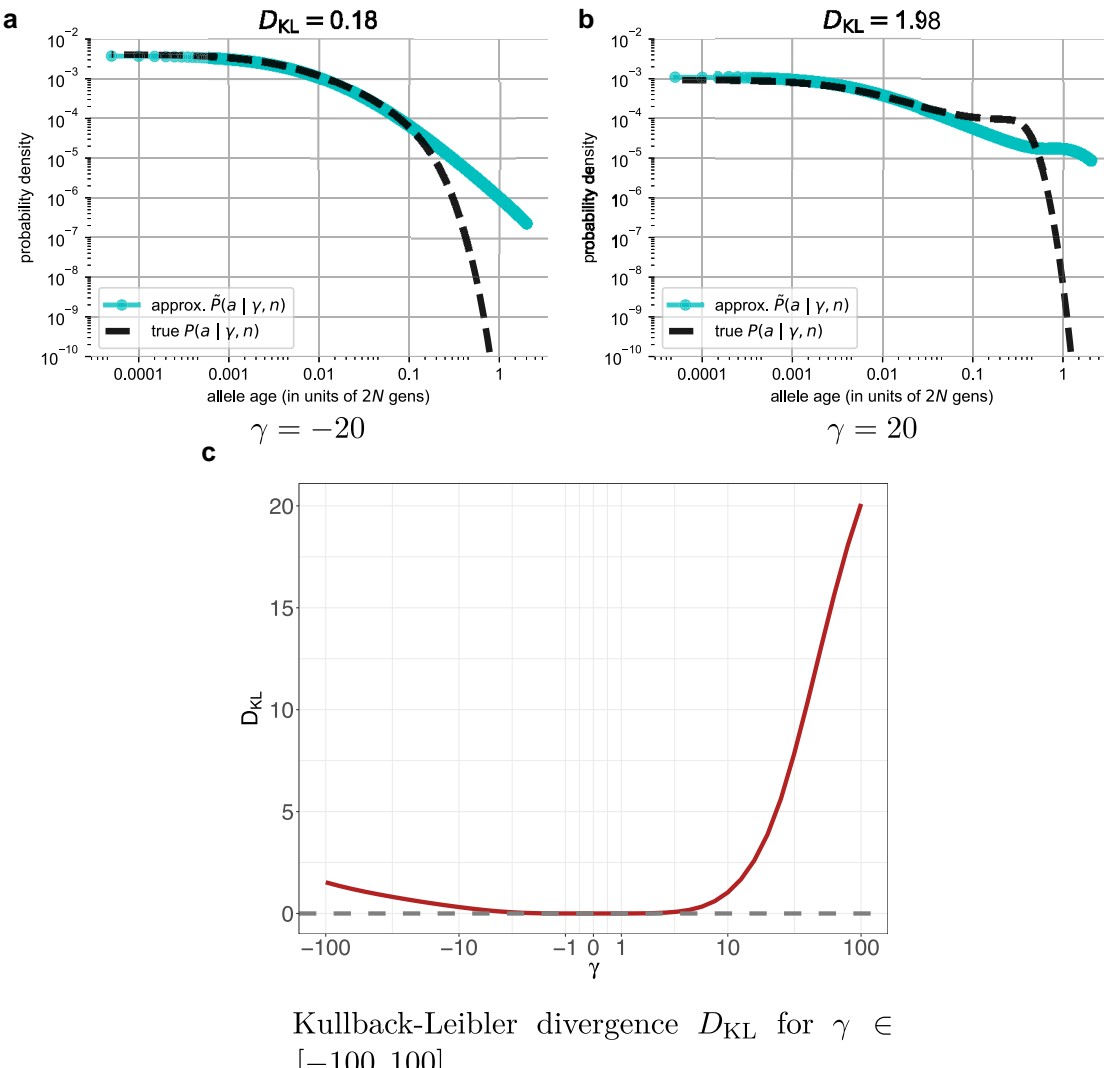

**Fig. 3.** The KL divergence between the unconditional age distributions of a particular selection coefficient ('true', Equation (10)) and the age distribution approximated by resampling from the neutral frequency spectrum ('approx.', Equation (9)) across a range of scaled selection coefficients. In a), we observe that for moderate negative selection ($\gamma = -20$), the two distributions are very similar for young alleles and differ only for the oldest alleles of which there are very few. In b), for sites experiencing moderate positive selection ($\gamma = 20$) the agreement between the true and approximated age distribution is much worse than for negative selection, particularly . In c), we plot the KL divergence between the true and approximated age distribution (Equation (7)) across a range of selection coefficients (see Supplementary Fig. S9 for an analog of this plot with total variation distance). a) $\gamma = -20$, b) $\gamma = 20$, and c) KL divergence $D_{KL}$ for $\gamma \in [-100, 100]$.

$$D_{KL}\big(P(a \mid \gamma, n) \,\|\, \tilde{P}(a \mid \gamma, n)\big) = \sum_{a=1}^{T_{max}} P(a \mid \gamma, n) \times \log_{10}\left(\frac{P(a \mid \gamma, n)}{\tilde{P}(a \mid \gamma, n)}\right). \quad (11)$$

In Fig. 3c, we plot this divergence as a function of the scaled selection coefficient for $\gamma$ ranging from −100 to 100 (see also Supplementary Fig. S9, where we consider an alternative measure, the total variation distance). In general, we see that the divergence is very close to zero for weakly selected alleles, whether they are under positive or negative selection. The divergence ultimately begins to increase for alleles under strong negative selection, so that with $\gamma = -100$, $D_{KL} = 1.53$, indicating that the ages are $10^{1.53} \approx 34$ times more likely under the true age distribution than the neutral resampling approximation. Under equally strong positive selection, in contrast, the divergence explodes as $\gamma$ increases, with $D_{KL} = 20.1$ when $\gamma = 100$. In this case, the ages are approx. $10^{20}$ times more likely under the true age distribution, reflecting the

unexpectedly recent origins of high frequency alleles under the true distribution relative to the neutral resampling approximation.

## Utility of allele ages in DFE inference

We now turn our focus to directly assessing the utility of allele age estimates for estimating the strength of negative selection acting on a variant or set of variants. Our results in the previous section show that for negatively selected alleles, once we condition on the distribution of allele frequencies, the distribution of ages is relatively insensitive to the precise value of the selection coefficient. This suggests that once the frequency of a variant under negative selection is measured, its age is unlikely to contain much additional information about its selection coefficient. To test this prediction, we used our numerical framework to implement simple inference frameworks for estimating selection coefficients using (1) only the SFS, or (2) the joint distribution of site frequencies

and ages. Both implementations use the standard PRF model introduced by Sawyer and Hartl (1992) to compute the likelihood of the selection coefficient (see Methods).

To compare the two approaches, we performed an experiment in which we simulated paired allele frequencies and ages for 1,000 unlinked sites under additive selection using `PReFerSim` (Ortega-Del Vecchyo *et al.* 2016) across a grid of scaled selection coefficients ranging from $\gamma = -100$ to $\gamma = 100$. We then inferred the value of $\gamma$ via maximum likelihood using both frequency-only and frequency & age. For each value of $\gamma$, we replicated this procedure 100 times, and visualized the distribution of MLEs across these 100 replicates in Fig. 4 (also see Supplementary Table S1b). The estimates from both approaches are largely unbiased, with the exception of a slight downward bias for the frequency-only approach in the case of weak positive selection. To measure how

much adding the ages to the estimation increases accuracy, we divide the variance of the MLEs derived from the frequencies-only by the variance of the MLEs derived from the joint frequency and age data. Notably, the variance of an MLE is asymptotically equal to the inverse of the Fisher information, which we can also compute directly with our numerical framework using finite differences (see Methods). We find good agreement between these two approaches. We include both versions in Fig. 4c, but rely largely on the simulation based approximation going forward, because it is much faster to compute.

We find that even when the allele ages are known exactly, including them in the inference results in only a small decrease in the variance of the estimators relative to the frequency-only approach for negatively selected alleles. This is consistent with our predictions above, and in contrast to positively selected alleles,

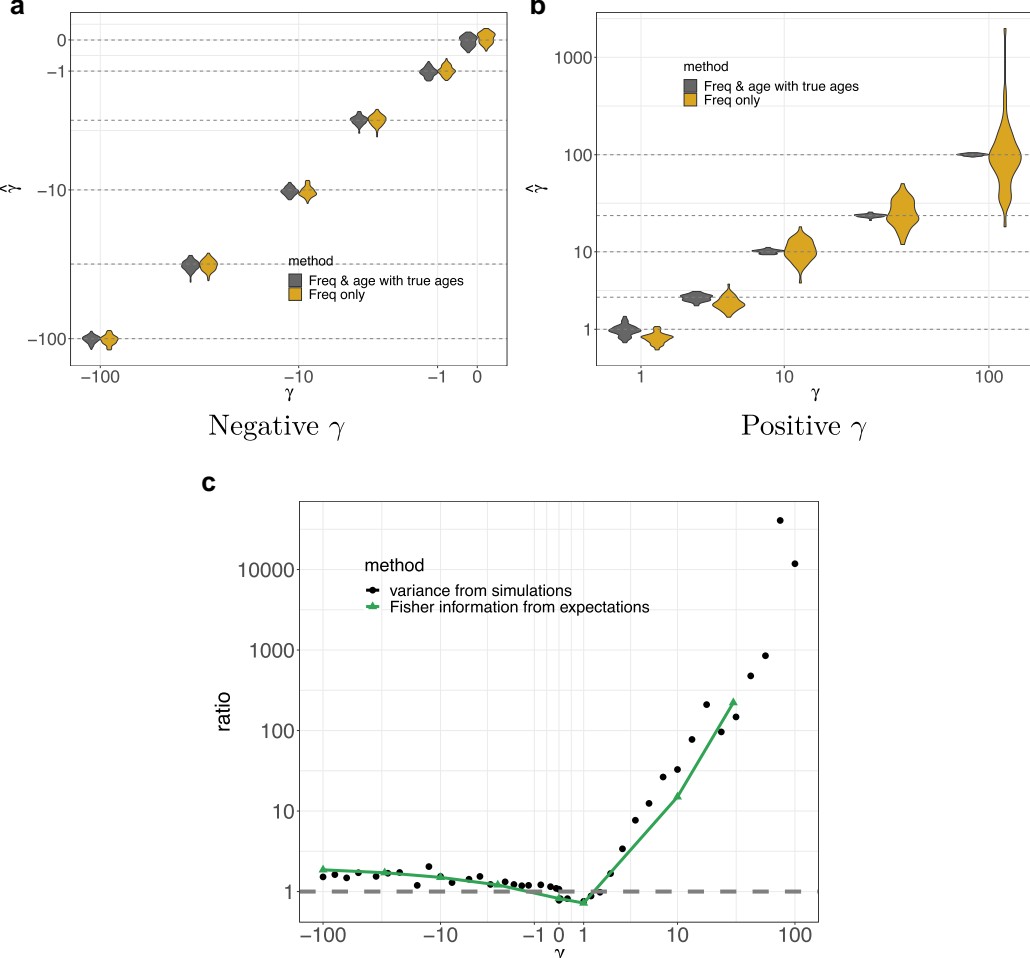

The observed ratio of variances across all $\gamma$ and the expected ratio of Fisher information metrics across selected values of $\gamma$

**Fig. 4.** Selection coefficient estimation for a constant demographic history of $N = 10,000$ using data simulated with `PReFerSim` (Ortega-Del Vecchyo *et al.* 2016) for a sample of $n = 100$. a,b) Violin plots showing accuracy of estimation for different values of population-scaled selection coefficient $\gamma$ using allele frequency & age data versus allele frequency alone. The X-axis shows different values of simulated $\gamma$, while the Y-axis shows the distribution on estimated $\hat{\gamma}$ over 100 independent replicates. The dashed horizontal lines denote the simulated values to aid in visualizing bias. On the negative side of the spectrum, we found that the MLE are close to the true value in both cases, with the approach including ages having slightly smaller error bars indicating more information about the selection coefficient in the data (especially for stronger values of selection). c) The ratio of variance (squared standard error) estimates (shown in black circles) calculated using Equation (19) from the frequency-only approach and the frequency & age approach for $\gamma \in [-100, 100]$. This tracks very closely with the expected ratio of Fisher information metrics (shown in green triangles, Equation (18)) for selected values of $\gamma$ across the range. a) Negative $\gamma$, b) Positive $\gamma$, and c) The observed ratio of variances across all $\gamma$ and the expected ratio of Fisher information metrics across selected values of $\gamma$.

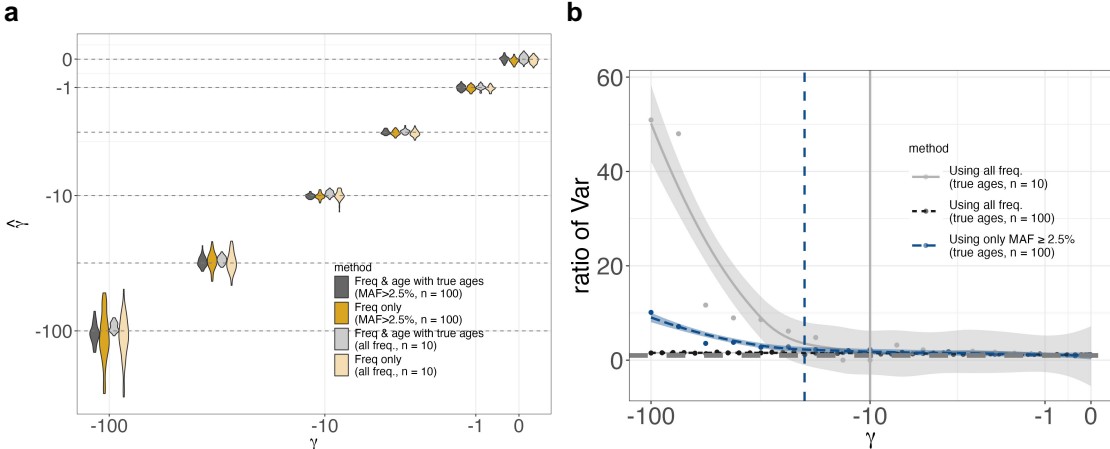

**Fig. 5.** Selection coefficient estimation for neutrality and negative selection for a constant demographic history of $N = 10,000$ using data simulated with `PReFerSim` (Ortega-Del Vecchyo *et al.* 2016) under two different schemes: large sample of $n = 100$ but only using sites with MAF ≥ 2.5% and a small sample of $n = 10$ but using *all* segregating sites. a) Violin plots showing the distribution of estimates across replicates under the two schemes, with or without ages. Estimates from both approaches (and both schemes) are similarly unbiased across the entire tested range. b) Ratio of variances and a loess fit (similar to Fig. 4c) to illustrate the gain in information due to including ages when there is a threshold on the SFS (with a sample of $n = 10$, observing a singleton is akin to imposing an MAF ≥ 5% threshold). In the case with the larger sample size, we observe a significant increase in information gain (compared with observing all sites) for $|\gamma| > 1/(2 \times 0.025) = 20$ (indicated by the dashed line). Similarly, in the case of the smaller sample, conditioning on segregation (i.e. observing 1/20) is the same as applying an MAF threshold, we see a significant increase in information gain for $|\gamma| > 1/2 \times 0.05 = 10$ (indicated by the solid line).

where ages provide a significant benefit (Fig. 4c). We observed a similar pattern when both the simulations and the inference are performed under the Tennessen *et al.* (2012) piecewise constant model of exponential growth (Supplementary Figs. S11 and S12).

In Fig. 3, we showed that age distributions conditioned on sample frequency differ significantly from neutral expectations for variants above a threshold frequency that depends on the scaled selection coefficient. Given this observation, we hypothesized that allele ages would be of greater benefit in samples with a threshold on the SFS (as is the case for genome-wide significant GWAS associations). We tested this prediction by inferring selection coefficients from simulated data with a minor allele frequency (MAF) cutoff of $x^\star = 0.025$ in a sample of $n = 100$. For scaled selection, coefficients greater than one over twice the frequency threshold (i.e. $\gamma > 1/2x^\star$) including ages in the inferences results in a sizeable increase in accuracy, but had little effect for scaled selection coefficients below this threshold (Fig. 5b).

This result also has implications for the use of ages in small samples. We can think of taking a sample of size $n$ and considering all sample frequencies as similar to imposing a threshold on the population allele frequency. For alleles under strong negative selection with $\gamma \gg n$, most alleles that segregate in the population will be absent from the sample, because they are too rare. Those that are included in the sample will be the ones that by chance have drifted up to population frequencies on the order of $1/2\gamma$, and will be represented in the sample predominantly as singletons. Thus, because we expect little frequency variation within the sample, this situation is very similar to imposing a threshold on the sample frequency within a larger sample, and suggests that the ages should provide additional information in this setting. To illustrate this, we repeat our simulations from above, but this time in a sample of $n = 10$, with no additional threshold on the sample frequency (akin to saying $x^\star = 0.05$). Consistent with our expectations, we observe that ages provide little utility in small samples when $\gamma \lesssim 1/2x^\star = 1/0.1$, but become valuable above this threshold (Fig. 5b). However, because most sites with such strong selection coefficients will not be included in the sample at all, we

would still expect small samples to be limited in their utility for learning about strong selection.

In practice, ages are typically estimated from haplotypic data under a neutral prior. Conditional on the inferred tree, the posterior density on a mutation's age then is uniformly distributed along the branch on which it arises. This leads to uncertainty about the exact age of the allele, as well as an upward bias for selected alleles (Maruyama 1974; Kiezun *et al.* 2013). To understand how these factors would impact inference of negative selection coefficients, we used `mssel` (Hudson 2002) to simulate haplotypic data consistent with negative selection acting at a focal site. We then used `Relate` (Speidel *et al.* 2019) to estimate a single gene tree for each simulated focal site, and then used the allele ages implied by these trees to estimate selection coefficients. In the simplest method, we used as a point estimate the midpoint of the branch containing the focal mutation. This estimate is the posterior mean under a neutral prior, conditional on the estimated tree, and unsurprisingly results in selection coefficient estimates that are biased toward zero (Fig. 6a). To better account for uncertainty in the estimated age, we next considered averaging over the uniform (neutral) posterior density on age on the branch on which the mutation arose. Interestingly, this approach was sufficient to remove the bias in the selection coefficient estimates, even though the age estimates still rely on a neutral prior (see Fig. 6a). However, the added uncertainty about when the mutation arose is sufficient to abrogate what little improvement in accuracy the ages had provided when the full SFS is observed (Fig. 6b).

Consistent with our observations for true ages, when there is a threshold on the SFS such that only sample frequencies greater than $x^\star = 0.025$ are observed, *and* the ages are estimated, then using the ages does result in a substantial decrease in the standard error of the estimated selection coefficients (Fig. 6c). However, in this case, averaging the inference over a uniform distribution on allele ages given the inferred tree does not fully remove the downward bias in the estimated selection coefficients, particularly when the scaled selection coefficient is large relative to the frequency threshold (i.e. $\gamma \gg 1/2x^\star$; Fig. 6d). Intuitively, this

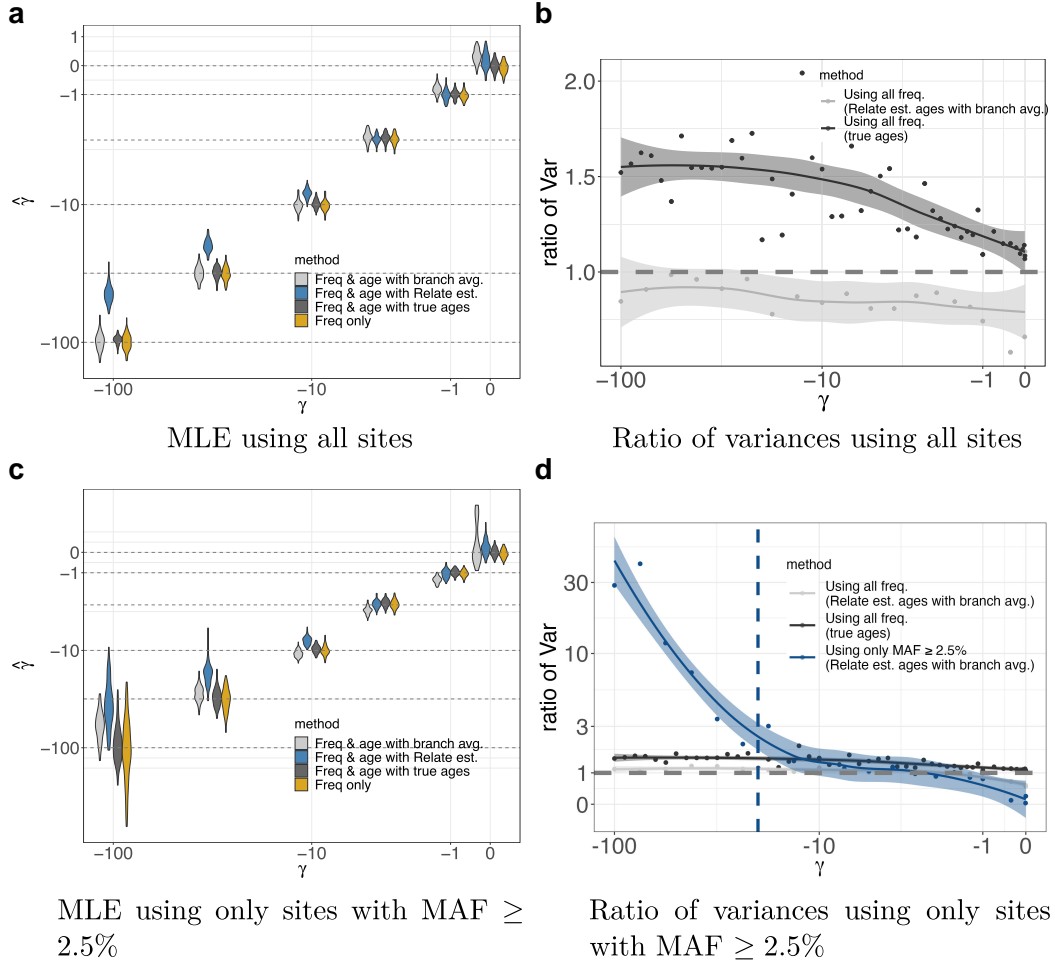

**Fig. 6.** Selection coefficient estimation accuracy for neutrality and negative selection under a constant demographic history with $N = 10{,}000$ using data simulated using `PReFerSim` (Ortega-Del Vecchyo *et al.* 2016) and haplotypes simulated using `mssel` (Hudson 2002), and ages estimated using `Relate` (Speidel *et al.* 2019) for a sample of $n = 100$. a) Using sites at all frequencies and raw age estimates from `Relate`, estimated selection coefficient are biased toward zero due to the neutral coalescent prior. However, this bias is eliminated by averaging over the density on age on the branch which it arose versus using the point estimate from `Relate`. b) However, uncertainty in the age estimates (i.e. increase in length of estimated branches) also erases nearly all of the additional information gained by including age estimates. c) When using the SFS thresholded at MAF ≥ 2.5%, estimates using the age density averages across the estimated branch also become biased, especially for larger scaled selection coefficients. d) Despite the bias seen in panel c, including ages still substantially reduces the variance of the estimates when the SFS is thresholded and ages are estimated using `Relate`. a) MLE using all sites, b) Ratio of variances using all sites, c) MLE using only sites with MAF ≥ 2.5%, and d) Ratio of variances using only sites with MAF ≥ 2.5%.

is because the sites included in the inference, which all have frequencies just above the threshold frequency $x^\star$, are precisely the ones for which the true trees deviate most significantly from the expectation under the neutral prior. Fully accounting for this effect would require an importance sampling scheme that averages over trees while upweighting contributions from those with younger allele ages (Coop and Griffiths 2004; Stern *et al.* 2019; Vaughn and Nielsen 2024). Such an approach could be implemented in our sample based scheme (see Discussion), but we do not pursue it here.

## Methods
### PRF model

The standard PRF model for the SFS as introduced in Sawyer and Hartl (1992) is given by

$$X_i \sim \mathrm{Pois}\big(\Phi^0_{2n}[i \mid \gamma]\big), \tag{12}$$

where $X_i$ is the number of sites observed with sample frequency $i$ and $\Phi^0_{2n}[i \mid \gamma]$ is the expected number of sites at sample frequency $i$

in the present generation given the scaled selection coefficient $\gamma$. We note that (1) elsewhere in the manuscript, we suppress this conditioning in our notation, writing just $\Phi^0_{2n}[i]$, but here we make it explicit so that its role in the inference is clear, and (2) although we write this section in terms of the *constant size* (equilibrium) case for simplicity, the extension to the nonequilibrium case is straightforward.

This framework can be expanded to the joint distribution of frequency and age via a simple Poisson splitting argument. If we write $X_{ia}$ for the number of sites observed to have frequency $i$ and age $a$, then the distribution of $X_{ia}$ arises from splitting the $X_i$ across age bins, so that

$$X_{ia} \sim \mathrm{Pois}\Big(\Phi^{0|a}_{2n}[i \mid \gamma]\Big), \tag{13}$$

where $\Phi^{0|a}_{2n}[i \mid \gamma] = \Phi^0_{2n}[i \mid \gamma]P(a_\ell \mid i_\ell, \gamma, n)$ is the expected number of such sites given a selection coefficient $\gamma$ (see also Supplementary Section S1, where we provide a method to compute $\Phi^{0|a}_{2n}[i \mid \gamma]$ directly). The likelihood of the selection coefficient can thus be written

$$\mathcal{L}(\gamma; \mathbf{X}) = \prod_{i=1}^{2n-1} \prod_{a=1}^{T_{\max}} P\left(X_{ia} \mid \lambda = \Phi_{2n}^{0|a}[i \mid \gamma]\right), \quad (14)$$

where $P(X_{ia} \mid \lambda = \Phi_{2n}^{0|a}[i \mid \gamma])$ is the Poisson probability of $X_{ia}$ given $\Phi_{2n}^{0|a}[i \mid \gamma]$. Alternatively, we can also rewrite (14) explicitly in terms of the Poisson splitting argument for the $X_i$, as

$$\mathcal{L}(\gamma; \mathbf{X}) = \prod_{i=1}^{2n-1} P\left(X_i \mid \lambda = \Phi_{2n}^{0}[i \mid \gamma]\right) \prod_{\ell \in \mathcal{S}_i} P(a_\ell \mid i_\ell, \gamma, n), \quad (15)$$

where $\mathcal{S}_i$ indicates the set of sites found at frequency $i$ in the sample.

In practice, existing gene tree estimation methods (e.g. `Relate` from Speidel *et al.* (2019), `GEVA` from Albers and McVean (2020), `tsdate` from Wohns *et al.* 2022, etc.) employ a neutral coalescent prior, and thus provide biased age estimates for alleles under selection. To counteract this effect, we apply a branch averaging scheme which integrates over the probability of observing an allele with a particular frequency and age given by the duration of the entire branch on which it arose.

Concretely, for a given site $\ell$ (and a tree estimated via the corresponding haplotypes), we draw $D$ mutations (each indexed by $d$) uniformly across the branch on which the allele arose

$$\hat{a}_\ell^{(d)} \mid \gamma = 0, \hat{a}_{\ell,b}, \hat{a}_{\ell,e} \sim \text{Unif}(\hat{a}_{\ell,b}, \hat{a}_{\ell,e}),$$

where $\hat{a}_{\ell,b}$ is the most recent generation in which the branch exists, and $\hat{a}_{\ell,e}$ the most ancient generation in which it exists.

Now, our likelihood for the selection coefficient includes an integration over the density on age given a selection coefficient and a branch (i.e. tree),

$$\mathcal{L}(\gamma; \mathbf{X}) \approx \prod_{i=1}^{2n-1} P\left(X_i \mid \lambda = \Phi_{2n}^{0}[i \mid \gamma]\right) \prod_{\ell \in \mathcal{S}_i} \frac{1}{D} \sum_{d=1}^{D} P\left(\hat{a}_\ell^{(d)} \mid i_\ell, \gamma, n\right). \quad (16)$$

This equation differs from Equation (15) in that it simply replaces the probability of observing a single value with a distribution over a set of values.

## Simulation and estimation framework

The goal of our simulations was to produce paired data of allele frequency & age $\{i_\ell, a_\ell\}$ for a site $\ell$ given a population-scaled selection coefficient $\gamma = 2Ns$ (assuming additive selection, $h = 1/2$) and demographic history $\{N\}$. We used the forward-in-time simulator `PReFerSim` to generate unlinked sites under each scenario due to its speed and its ability to record ages and trajectories of selected alleles (Ortega-Del Vecchyo *et al.* 2016). We tested the model under two demographic scenarios: a *constant size* case, where we set $N = 10,000$ for 100,000 generations (sufficient to reach approximate equilibrium for the entire range of simulated selection coefficients), and a *piecewise exponential growth* case inferred in Tennessen *et al.* (2012) for humans over the last 50,000 generations ("Africa_1T12" from Adrion *et al.* 2020, see Supplementary Fig. S2). To capture the entire range of selection strengths, we simulated under 30 different selection coefficients, $s$, equally spaced on a log-scale from $5 \times 10^{-7}$ (very weak, corresponds to $\gamma = 0.01$ in the *constant size* case) to $5 \times 10^{-3}$ (strong, corresponds to $\gamma = 100$ in the *constant size* case), and $s = 0$. This was repeated on both sides of the spectrum. The population scaled mutation rate $\theta$ was varied from 400 for $s = -5 \times 10^{-3}$ to 10 for $s = 5 \times 10^{-3}$, so as to

obtain a similar number of segregating sites ($\approx 1,000$) across all selection coefficients. This was to ensure that the information about the selection coefficient came from the frequency and age, and not from the total number of segregating sites in the data.

Secondly, to explore the performance of the model in the case of biased (or estimated) ages, we used `PReFerSim` to record entire trajectories of alleles, which was used as input to the program `mssel` (Hudson 2002) that outputs haplotypes containing the selected allele. These haplotypes were passed into `Relate` (Speidel *et al.* 2019) to construct gene trees at the selected sites and subsequently *estimate* ages of the selected mutations. To understand the extent of the bias under these estimated ages, we used the midpoint of the corresponding branch in the selection coefficient estimation framework instead of the true ages from before. To mitigate this bias for alleles under negative selection, we used the likelihood in Equation (16) with $D = 500$ mutations uniformly on the branch on which the mutation arose, as we found that the estimate of the selection coefficient did not change significantly with a higher number of draws. This exploration was done only for the *constant size* case, as we expect the findings to extend to more complex demographic histories.

For each combination of selection coefficient and population size history, we ran 100 replicates to get an accurate measure of the first two moments (mean and variance) of the distribution of the estimated selection coefficient, $\hat{s}$. In all cases, we used a maximum-likelihood based approach to estimate the selection coefficients. Since we were optimizing over a single dimension, we used Brent's method (default) in `scipy` (Virtanen *et al.* 2020) to minimize the negative log-likelihood of the data under the appropriate models. This could be extended to estimate multiple selection parameters (for instance, shape and scale of a gamma DFE).

### Measuring the information content of allele ages

To quantify the extent to which adding allele ages improves our estimates of $\gamma$, we compute the ratio of the Fisher information metrics between the frequency & age method and the frequency-only method, i.e.

$$\frac{\mathcal{I}_{i,a}(\gamma)}{\mathcal{I}_i(\gamma)} \quad (17)$$

where

$$\mathcal{I}_i(\gamma) = \sum_{i=1}^{2n-1} \left(\frac{\partial}{\partial \gamma} \log P(i \mid \gamma, n)\right)^2 P(i \mid \gamma, n)$$

$$\mathcal{I}_{i,a}(\gamma) = \sum_{a=1}^{T_{\max}} \sum_{i=1}^{2n-1} \left(\frac{\partial}{\partial \gamma} \log P(a \mid i, \gamma, n) P(i \mid \gamma, n)\right)^2 P(a \mid i, \gamma, n) P(i \mid \gamma, n),$$

$$(18)$$

and we compute the derivatives using finite difference methods. The higher this ratio, the more peaked the likelihood surface becomes when we add allele ages, relative to the frequency-only inference.

Because computing these derivatives was fairly time consuming, we also approximate the ratio in (17) by computing the variance of the MLE across 100 simulation replicates. We know from the Cramér–Rao bound (Cramér 1999; Rao 1992) that

$$\text{Var}(\hat{\gamma}) \geq \frac{1}{\mathcal{I}(\gamma)},$$

where the equality holds asymptotically for maximum-likelihood estimators, as is the case for us. Thus, in the limit of a large number of sites, we expect that

$$\frac{\mathcal{I}_{i,a}(\gamma)}{\mathcal{I}_i(\gamma)} \approx \frac{\mathrm{Var}(\hat{\gamma} \mid i)}{\mathrm{Var}(\hat{\gamma} \mid i, a)} = \frac{\frac{1}{99} \sum_{j=1}^{100} (\hat{\gamma}_j^i - \bar{\gamma}^i)^2}{\frac{1}{99} \sum_{j=1}^{100} (\hat{\gamma}_j^{i,a} - \bar{\gamma}^{i,a})^2} \qquad (19)$$

where $\hat{\gamma}_j^i$ denotes an estimate of $\gamma$ obtained from frequency-only data (i.e. Equation (12)) for the $j$th simulation replicate, $\hat{\gamma}^{i,a}$ denotes an estimate obtained from frequency & age (i.e. Equation (13)), and $\bar{\gamma}$ refers to the mean MLE across replicates. We present a comparison of these quantities in Fig. 4c for the case with a constant size population.

## Discussion

Here we develop efficient numerical methods for computing the distribution of ages for selected alleles. This method improves on prior approaches to this problem by avoiding the computational cost and Monte Carlo error associated with simulations, producing accurate numerical approximations in seconds. Our work also builds on previous methods which have aimed to use the information about allele ages contained within patterns of haplotype variation to learn about the distribution of selection coefficients acting on negatively selected alleles (Kiezun *et al.* 2013; Johri *et al.* 2020; Ortega-Del Vecchyo *et al.* 2022). Interestingly, we find that if all bins of the SFS are observed in a reasonably large sample, then allele ages provide relatively little additional information about negative selection coefficients, particularly when we account for the fact that ages are estimated with error. However, incorporating ages can provide larger benefits if the frequency spectrum is truncated, e.g. in statistical genetics analyses of "common variants" (e.g. Gazal *et al.* 2017; Kichaev *et al.* 2019; Simons *et al.* 2022). Notably, because our method separates the problem of inferring allele ages from the problem of learning about selection coefficients conditional on the inferred allele ages, our results capture fundamental limits which cannot be overcome via improvements to the methods for inferring ages.

Nonetheless, there are several ways that our methods could be extended or improved upon. For example, although we focus on models with only a single population, the `moments` framework can accommodate multiple populations related via ancestral population splits, admixture events, and continuous migration. Computing the distribution of allele ages conditional on the sample frequencies observed across multiple populations should therefore be relatively straightforward, merely requiring additional bookkeeping to account for which population the allele ultimate arose in. Another plausible extension would involve replacing the Wright–Fisher diffusion by the discrete time Wright–Fisher model using methods recently developed by Spence *et al.* (2023). This approach could be used either to obtain the distribution on age given the population frequency, or by subsampling from this population SFS, the sample frequency that we focus on here. This approach would have the benefit that it would not rely on the assumption inherent to the Wright–Fisher diffusion that there is at most one coalescent event in the history of the sample per generation, making it more amenable to stronger selection and very large samples, at the cost of the increased computational expense required to track the full population and to compute a broader transition distribution in each generation.

Another way of improving upon our method would be to incorporate additional information about the frequency trajectory beyond that contained in just the age of the allele. Several such methods have been developed in the context of both temporal sampling/ancient DNA (Bollback *et al.* 2008; Malaspinas *et al.* 2012; Mathieson and McVean 2013; Irving-Pease *et al.* 2024; Mathieson and Terhorst 2022), or coalescent inference (Stern *et al.* 2019), or both (Vaughn and Nielsen 2024). We expect that it would be possible to extend our method in either of these directions. The possibility of extending our method to full coalescent inference is particularly interesting. Specifically, the expressions required for sampling from the ancestral selection graph (ASG, Neuhauser and Krone 1997) conditional on the present-day sample configuration in a population at demographic equilibrium depend on ratios of the stationary sampling probabilities of the Wright–Fisher diffusion (Stephens and Donnelly 2003). Despite the elegance of this approach, the need to simulate many lineages in the ASG which are not ancestral to the sample makes it computationally burdensome relative to the state-of-the-art structured coalescent HMM method (Stern *et al.* 2019; Vaughn and Nielsen 2024), which explicitly models the latent population allele frequency. The requirement that these "virtual" lineages be simulated to obtain a valid sample stems from the fact that the outcome of a given selection event depends on the current frequency of the allele in the population, and thus cannot be determined until the outcomes of mutation and selection events occurring at earlier time points are known (Slade 2000). However, in preliminary work in this direction, we have found that if the stationary sampling probabilities can be replaced with time-varying sampling probabilities which have already accounted for the distribution of times at which the mutation could have arisen (i.e. the distribution of allele ages), then the need to simulate virtual lineages can be avoided. Alternatively, it may be possible to compute the probability of a genealogy under selection via a purely forward-in-time approach, using a combination of the Moran model framework implemented in `momi` (Kamm *et al.* 2017) and the jackknife approximated selection operator introduced by Jouganous *et al.* (2017). Whether either of these approaches would offer benefits relative to the structured coalescent HMM method is an interesting question for future work.

However, given our focus on negative selection in this paper, it is worth asking whether additional information about the frequency trajectory beyond that already contained in the present-day frequency and the age (e.g. frequencies at intermediate time points), to contain much additional information about the strength of negative selection experienced by an allele. Conditional on the end-points of a frequency trajectory, a larger selection coefficient means that a larger fraction of the frequency change required to go from one end-point to the other occurs closer to the present (Schraiber *et al.* 2013). This, in turn, would lead to coalescent events on the derived background being shifted closer to the present. However, a larger selection coefficient also means that segregating deleterious alleles will be younger on average (Kimura and Ohta 1973; Maruyama 1974; Kiezun *et al.* 2013), so there is less time for selection to substantially alter the frequency trajectories. Put differently, conditional on segregation, most deleterious alleles exist at frequencies below the $x_\gamma^\star = 1/2\gamma$ threshold at which the frequency spectrum transitions away from neutral behavior, so their trajectories should look approximately neutral. We would therefore expect to gain little additional information from their genealogies beyond what is contained in the two end-points. However, similar to the case with the ages, if we restricted the sample to sites with frequencies above this threshold, then we might expect the genealogies, or other information about the frequency trajectory, to be more useful.

Another setting where our approach may have greater utility is in the inference of selection on transposable elements (TEs). For example, using neutral coalescent theory (Blumenstiel *et al.* 2014) and simulations (Horvath *et al.* 2022), prior work has shown that using an "age-adjusted" SFS (essentially, a binned version of the age conditioned SFS that we consider in Supplementary Section S1) could be beneficial for estimating selection coefficients in TE families where the impact of selection is confounded by time-inhomogenous bursts of replication. Although we do not explore this direction in this paper, time-varying rates can be readily incorporated into our framework. More generally, our work illustrates how the now well-established framework of recursions for the SFS can be leveraged to address fundamental questions in population genetic inference.

## Data availability

All data utilized in this study were generated through computational simulations. The simulation scripts and source code are available in this repository: https://github.com/VivaswatS/selCoefEst.git. The source code for the program `mssel` is available at https://github.com/dortegadelv/HaplotypeDFEStandingVariation/tree/master/Programs/Mssel.

Supplemental material available at GENETICS online.

## Acknowledgments

We would like to thank members of the Berg, Novembre, and Steinrücken labs, as well as members of the University of Chicago Program in Computational Biology (PCB) community for helpful discussions and feedback during the development of this project. We thank Aaron Ragsdale and Diego Ortega-Del Vecchio for help with implementation and simulation. Additionally, we thank John Novembre, Matthias Steinrücken, and Xuanyao Liu for support at all stages of this work, and Carl Veller and Yuval Simons for comments on the manuscript. Computing was performed on servers maintained by the University of Chicago Research Computing Center.

## Funding

This work was supported by NIH MIRA grant R35GM151257 to J.J.B.

## Conflicts of interest

The author(s) declare no conflicts of interest.

## Author contributions

V.S. and J.J.B. formulated the research question together, and developed the methods together. V.S. implemented the methods, ran the simulations, and wrote a first draft of the manuscript. The final manuscript was edited collaboratively by V.S. and J.J.B.

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

*Editor: K. Lohse*