## [Peer Review File · Genetics]

Allele ages provide limited information about the strength of negative selection

Vivaswat Shastry and Jeremy Berg

NOTE: The reviews and decision letters are unedited and appear as submitted by the reviewers.

In extremely rare instances and as determined by a Senior Editor or the EIC, portions of a review may be redacted. If a review is signed, the reviewer has agreed to no longer remain anonymous.

The review history appears in chronological order.

Review Timeline:

Submission Date:	2024-08-19
Editorial Decision:	2024-09-24
Resubmission Received:	2024-12-11
Accepted:	2024-12-12

September 24, 2024

GENETICS-2024-307352

Allele ages provide limited information about the strength of negative selection

Dear Dr. Berg:

Two experts in the field have reviewed your manuscript, and I have read it as well. Both reviewers are broadly positive about your study. While your manuscript is not currently acceptable for publication in GENETICS, we would welcome a revised manuscript. Both reviewers have made some very constructive comments and concerns to be addressed in a revised manuscript. You can read their reviews at the end of this email. Both reviewers emphasize the importance of your negative result that allele age does not provide (much) additional information about -ve selection even under ideal conditions; and I agree that this is worth highlighting. However, I also agree with the suggestion of reviewer 3 that it would be good to include an example of how your new method for efficiently computing $P(a|i,s)$ can be successfully employed to learn something new about evolution.

We look forward to receiving your revised manuscript. Please let the editorial office know approximately how long you expect to need for revisions.

Upon resubmission, please include:

1. A clean version of your manuscript;
2. A marked version of your manuscript in which you highlight significant revisions carried out in response to the major points raised by the editor/reviewers (track changes is acceptable if preferred);
3. A detailed response to the editor's/reviewers' feedback and to the concerns listed above. Please reference line numbers in this response to aid the editor and reviewers.

Your paper will likely be sent back out for review.

Additionally, please ensure that your resubmission is formatted for GENETICS
<https://academic.oup.com/genetics/pages/general-instructions>

Follow this link to submit the revised manuscript: Link Not Available

Sincerely,

Konrad Lohse
Associate Editor
GENETICS

Approved by:
Nicholas Barton
Senior Editor
GENETICS

Reviewer #2 :

This paper studies how allele age can be used to infer selection coefficients. The paper starts with a technical contribution, which is a novel method for computing the distribution of the age of a variant conditional on knowing the strength of selection and the present-day frequency. The authors use this method to investigate whether knowing the age of a variant can improve estimation of its selected (dis)advantage. The main finding is actually negative: for variants that are under weak negative selection (i.e. the majority, presumably) allele age does not contribute much additional information, beyond what is already implied by the allele frequency. This is established by numerical experiments and simulation studies, where it is shown that the distribution of allele age under neutral and non-neutral models are largely similar, at least for alleles that are already at low frequency. For variants that are ascertained to already be at intermediate (>2.5% say) frequency, the dependence is somewhat stronger.

The manuscript is well written and easy to follow, and my opinion is generally favorable. Although the message is mainly negative, I believe it will be interesting and useful to the community. The technical innovations are a nice extension of the

moments framework, and I think they could have applications elsewhere. If substantial revision is contemplated, I'd argue for expanding the manuscript to include at least one example of how the new method for efficiently computing $P(a|i,s)$ can be successfully employed to learn something new about evolution.

General comments

- Thinking of other settings/problems where this method could be useful:
- One could be studying selection on standing variation, so that more variants are above the $1/(2*\gamma)$ threshold. To study this you could e.g. compare $P(a|i,s_0=\dots=s_{\{Tmax\}}=0)$ and $P(a|i,s_0=\dots=s_{\{Tmax/2\}}=x,s_{\{Tmax/2+1\}}=\dots=s_{\{Tmax\}}=0)$.
- The focus here is all about using age to improve estimates of s . A persistent question I had is whether this method can be used to give use better estimates of a if we already know s ? (This is more relevant for directional selection, see next comment.)
- The scope of the paper could be broadened to include positive selection, where the method should work better. For example, use the new machinery to give a distribution over the age of some well-known positive selection (LCT, say) and compare your results with earlier attempts to date these.
- Section 3.2.1: aren't you essentially just comparing the expected Fisher/Godambe information here? Computing it directly, instead of simulating data and fitting MLE, could be more accurate.
- Equation (7): could you provide some intuition/discussion for why this is the correct approximation? Naively, I'd have supposed that $\tilde{P}(a|\gamma) \propto \sum_i P(a|i,\gamma=0) P(i|\gamma)$, i.e. without division by $P(i|\gamma=0)$, would work, based on $P(a|i,\gamma) \approx P(a|i,\gamma=0)$ by (6).
- This is more a personal preference, but I think total variation would be a better way to express all the divergences since it is interpretable as a percentage. I have no intuition for what (e.g. Figure 3b) $d_{KL}=4.549$ means.

Minor comments

(there are a lot of gaps in the line numbers, so these coordinates are approximate)

- 130: Missing a period "variants To"
- ~156: "In this case, the μ_t are also the same" You assume this in the previous sentence; I think you mean "the m_t "?
- ~216: "sites with population frequencies above and below this threshold are represented in the sample" I couldn't figure out what this passage is supposed to mean, please clarify.
- ~314: "Poisson splitting argument" I think this is usually called Poisson thinning?

Reviewer #3 :

Shastri and Berg present an interesting and slightly provocative result: knowing allele ages does not provide much more information about selection coefficients than simply knowing frequency information. They show this by first building an algorithm to compute the age distribution under time varying population sizes (and in fact time varying mutation rates and selection coefficients). They then incorporate this into a Poisson random field framework for inference of selection coefficients, and find pretty minimal reductions in the sampling variance of the MLE (although reducing it by a factor of $\sim\sqrt{10}$ for the exponential population growth case, as shown in Fig S8 isn't nothing!)

If I understand, the intuition for this is that under negative selection, alleles below a frequency of $1/(4N_s)$ are effectively neutral, and so basically have a neutral age distribution, and alleles above a frequency of $1/(4N_s)$ are basically dead, so their ages don't matter. Motivated by this, they argue that, under negative selection, the selected age distribution is well approximated by taking the neutral age distribution and reweighing it by the selected SFS---i.e. dampening ages that contribute to frequencies that are unlikely under selection and enhancing ages that are more likely under selection. They find that this works pretty well for negative selection, although not so much for positive selection.

I think a very interesting aspect of how this paper is framed is that it is in terms of assuming one *knows* the age (up to some error), unlike a lot of past work on allele age that is focused primarily on inference of allele age. Of course this framing makes sense because we have ancient DNA and genome-wide genealogy inference. Because of course ages are still only estimated uncertainty, the claims that even knowing ages with no uncertainty doesn't really help bodes poorly for situations in which ages are estimated (and this is confirmed in the author's simulations using Relate).

This is a very nice and important paper, especially in the current context, and I think it does a good job of exploring confounding factors by also examining a case of exponential growth under a reasonable model. I only have some minor comments on the manuscript.

- 1) It took me a while to realize that the denominator of Equation 2 is in fact just $\sum_a m_a$. *It might be worth just pointing out that, of course, the SFS is the sum over all ages that contribute to frequency i .*
- 2) *Also, related to equation 2, is it surprising that the population size at each age does not enter as a "prior" in Equation 2? i.e. it seems like more mutations enter when there is a larger population.*
- 3) *Although the authors provide some intuition behind Equation 3, I assume they derived it from Equation 1 rather than conjured*

it, and it would be nice for the authors to detail that derivation in an Appendix.

4) It might be nice to plot a figure like S6 under the non-constant model, just to get some intuition about what happens under that model and how it changes things.

I prefer to sign my reviews. My name is Joshua Schraiber.

Associate Editor Comments:

It would be helpful to mention some of the commonly used implementations of PRF based DFE inference in the intro: e.g. DFEalpha (Keightley and Eyre-Walker 2007), polyDFE (Tataru et al. 2017), fastDFE (Sendrowski & Bataillon 2024) in the intro.

Perhaps I missed this, but it was not clear to me what n was assumed in the simulation test (figs 4 & 5). Was this the $n=100$ as in the analytic comparison in Fig. 3?

More generally, only large samples ($n \sim 100$) are considered. Since SFS based inference of the DFE for non-human data typically involve much smaller samples ($n \sim 10$, e.g. <https://doi.org/10.1534/genetics.116.188102>), it would be helpful to show the sample size dependence of the extra info contained in allele age by adding Results for $n=10$ to Fig 4&5.

Fig 3: Although this is given in the figure, I think would help to spell out in the legend that the blue and dashed lines correspond to approximated and true allele distribution. I also wonder whether it would be more useful to express the x axis in $2 N_e$ generations.

Please also consider the reviewer's suggestions that "total variation would be a better way than KL divergence since it is interpretable as a percentage"

Responses to reviewer comments

Below, we provide detailed responses to each of the reviewers' comments. Reviewer comments are shown in black and our responses are shown in blue. To ease review of changes to our manuscript, we have included a PDF of the manuscript with the additions highlighted in blue and the deletions marked in red and placed in the footnotes.

Two experts in the field have reviewed your manuscript, and I have read it as well. Both reviewers are broadly positive about your study. While your manuscript is not currently acceptable for publication in GENETICS, we would welcome a revised manuscript. Both reviewers have made some very constructive comments and concerns to be addressed in a revised manuscript. You can read their reviews at the end of this email. Both reviewers emphasize the importance of your negative result that allele age does not provide (much) additional information about -ve selection even under ideal conditions; and I agree that this is worth highlighting. However, I also agree with the suggestion of reviewer 3 that it would be good to include an example of how your new method for efficiently computing $P(a|i, s)$ can be successfully employed to learn something new about evolution.

We look forward to receiving your revised manuscript. Please let the editorial office know approximately how long you expect to need for revisions.

We would like to thank the reviewers, and the editor, for their detailed reading of our manuscript, and for their comments. We feel that the manuscript has been improved as a result, and appreciate the opportunity to submit our revision.

We have attempted to thoroughly address all of the reviewers' concerns. However, we elected not to pursue the suggestion to provide an additional application of the method to "learn something new about evolution". While we agree that there are many potential directions the work presented here could be extended, this is a potentially difficult and open ended directive. The reviewer has offered a few specific suggestions, but we think it unlikely that we would learn enough by pursuing these directions to justify reorganizing the paper and diluting its current message (we include some more detailed comments to that effect below). Other directions (e.g., extending the method to estimate positive selection coefficients from full genealogies) we think have strong potential, but will require much more work than is feasible in a revision.

The present manuscript considers the evolutionary process itself as the object of interest, and advances our understanding of how we can learn about the evolutionary process, and where the limits of our ability to do so are. We think that this is an important contribution in its own right, and it seems the reviewers agree that this is the case. With this in mind, we ask that our revised manuscript be considered without any additional application.

Reviewer #2

The manuscript is well written and easy to follow, and my opinion is generally favorable. Although the message is mainly negative, I believe it will be interesting and useful to the community. The technical innovations are a nice extension of the moments framework, and I think they could have applications elsewhere.

Thank you!

If substantial revision is contemplated, I'd argue for expanding the manuscript to include at least one example of how the new method for efficiently computing $P(a | i, s)$ can be successfully employed to learn something new about evolution.

See main comment to the editor above, and comments below.

General comments

- Thinking of other settings/problems where this method could be useful:
- One could be studying selection on standing variation, so that more variants are above the $1/(2 * \gamma)$ threshold. To study this you could e.g. compare $P(a|i, s_0 = \dots = s_{T_{max}} = 0)$ and $P(a|i, s_0 = \dots = s_{T_{max}/2} = x, s_{T_{max}/2+1} = \dots = s_{T_{max}} = 0)$.

We assume that the scenario the reviewer has in mind is one of *positive* selection on standing variation. (We are unsure of what biological scenario would motivate considering such a scenario for negative selection)

We share the reviewer's sentiment that understanding selection on standing variation is a broadly interesting question. Notably, the senior author of the present manuscript is also the author of prior work developing a model for a selective sweep of a standing variant (Berg and Coop 2015, Genetics, doi: 10.1534/genetics.115.178962).

The theory laid out in that paper could be used to develop a rough approximation for e.g. "the age distribution of a variant currently found at frequency x , which became beneficial with selection coefficient s , t generations ago".

In principle, we could also apply the numerical tools developed in our present paper to compute the exact age distribution under such a scenario. This exact distribution would have a higher variance than the one following from the Berg and Coop 2015 approximation, because it would incorporate extra sources of variation that the Berg and Coop 2015 theory ignores in order to arrive at approximation that were tractable at the time. However, it is not clear to us what we would learn from this that would make it interesting enough to reshape some of the paper's message around it.

The tools that we have developed here could be extended to study selection on standing variation, either by modeling time series data (similar to <https://doi.org/10.1101/2024.05.10.593575>), or coalescent trees (similar to <https://academic.oup.com/mbe/article/41/8/msae156/7724092>), and we think extending the moments framework in these directions could have benefits. But doing so is not trivial, and is beyond our scope for the current paper.

- The focus here is all about using age to improve estimates of s . A persistent question I had is whether this method can be used to give use better estimates of a if we already know s ? (This is more relevant for directional selection, see next comment.)
- The scope of the paper could be broadened to include positive selection, where the method should work better. For example, use the new machinery to give a distribution over the age of some well-known positive selection (LCT, say) and compare your results with earlier attempts to date these.

Yes, similar to above, the machinery developed here could be extended in this direction. However, here again, the immediate utility is not obvious to us. Notably, *both* the selection

coefficient and the allele age are unknowns, so we think that the optimal way to learn about them is via a joint inference, in which both are simultaneously treated as unknowns, and we maximize a joint likelihood or infer a joint posterior distribution. Similar to the previous comment, we think the methods developed here could be extended to jointly infer the selection coefficient and age (and/or other features of the frequency trajectory) from time series data and/or full coalescent trees, and we think such developments could have advantages, but are beyond our present scope.

Setting that possible direction aside, we could simply chose some values of s from the literature and plug them in and see what we get. However, while it is widely agreed that the LCT gene has been a target of strong positive selection in the recent past, there is currently considerable debate in the literature about when selection actually started acting on the variant, and what exactly the selection coefficient is. Without conducting a complete analysis of time series/coalescent trees ourselves, any inference about allele age using estimates of s from the literature would largely reflect assumptions we would have to make about which estimates are more plausible. We are skeptical that this would represent a valuable contribution to our understanding of the evolution of LCT.

- Section 3.2.1: aren't you essentially just comparing the expected Fisher/Godambe information here? Computing it directly, instead of simulating data and fitting MLE, could be more accurate.

True, the ratio we are computing is, asymptotically, equal to the ratio of the Fisher/Godambe information of the two different approaches. In the updated manuscript, we compute the Fisher/Godambe information exactly for a few points the first time that we use it (Figure 4), and find good agreement with the simulation based method. Given this, we retain the original simulations based method in later figures (notably, in Figure 6 when we consider using ages estimated from Relate, we must use simulations because we cannot compute the true sampling distribution of the Relate estimates).

- Equation (7): could you provide some intuition/discussion for why this is the correct approximation? Naively, I'd have supposed that $\tilde{P}(a|\gamma) \propto \sum_i P(a|i, \gamma = 0)P(i|\gamma)$, i.e. without division by $P(i|\gamma = 0)$, would work, based on $P(a|i, \gamma) \approx P(a|i, \gamma = 0)$ by (6).

Apologies for the confusion. We have rewritten this section of the text so that it is clearer. The main idea is that we are imagining resampling from the neutral distribution to obtain an SFS that matches that of selected alleles, but with the age distribution conditional on sample frequency that still matches neutrality. The ratio $\frac{P(i|\gamma)}{P(i|\gamma=0)}$ thus acts as a resampling weight.

The updated paragraph reads:

Intuitively, we imagine first sampling pairs of frequencies and ages for a large number of neutral alleles. We then imagine retaining each sampled site with probability proportional to

$$w_i = \frac{P(i | \gamma, n)}{P(i | \gamma = 0, n)} \quad (1)$$

where $P(i | \gamma, n)$ and $P(i | \gamma = 0, n)$ are the sample SFSs, conditional on segregation, for selected and neutral alleles respectively. The resulting distribution of allele frequencies among

the resampled sites will then be a valid sample from the distribution of frequencies under selection (that is, from $P(i | \gamma, n)$). Within each frequency bin, the ages distribution still follows the neutral distribution $P(a | i, \gamma = 0, n)$, but resampling the SFS to match the selected SFS largely suppresses the contributions from the frequency bins where the true age distribution is strongly impacted by selection. The approximate age distribution implied by this resampling procedure can be computed directly as

$$P(a | \gamma, n) \approx \tilde{P}(a | \gamma, n) = C^{-1} \sum_{i=1}^{2n-1} w_i P(a | i, \gamma = 0, n), \quad (2)$$

where $C = \sum_{i=1}^{2n-1} w_i$ is a re-normalization constant which ensures that $\sum_{a=1}^{T_{\max}} \tilde{P}(a | \gamma, n) = 1$. Figure S7 illustrates the resampling weights, w_i , as a function of i for a few different choices of γ .

- This is more a personal preference, but I think total variation would be a better way to express all the divergences since it is interpretable as a percentage. I have no intuition for what (e.g. Figure 3b) $d_{KL} = 4.549$ means.

We agree that our initial draft did not provide enough intuition for how to think about the values of the KL divergence. However, we still think it is the right measure for this case. Here, we will try to explain why. The KL divergence between distributions p and q is the expectation of the log-likelihood ratio between p and q for a single randomly chosen data point, assuming the data was generated from p . That is $D_{KL}(p || q) = \mathbb{E}_p \left[\log \frac{p}{q} \right]$. If we exponentiate the KL divergence, this gives the geometric mean of the log likelihood ratio for a single data point. And because likelihoods combine multiplicatively, this geometric mean represents the right way of assessing how much more likely the data is on average under p relative to q . For example, in our initial submission we expressed the KL divergence in terms base e . So, $D_{KL} = 4.549$ for $\gamma = 20$ in Figure 3B means that the ages of derived alleles at a segregating site with $\gamma = 20$ are $e^{4.549} \approx 94.5$ times more likely on average under the true age distribution than under the resampling approximation. In contrast, for $\gamma = -20$, $D_{KL} = 0.418$, so the ages are only $e^{0.418} \approx 1.52$ more likely on average under the true distribution than the resampling approximation.

We think that this interpretation is useful, because it speaks to the information content of the allele ages once one knows that the frequency matches the SFS of a particular selection coefficient, and thus helps explain why the ages are relatively uninformative for negatively selected alleles.

In order to facilitate this interpretation, we now explain this in the text the first time we use the KL divergence (lines 234-240). We also note that in order to facilitate the easy interpretation of the KL divergences as a measure of how much more likely the data are under the true distribution than the approximate one, we now express all of our KL divergences in base 10, and note this clearly in the text.

We think that this interpretation offered by the KL divergence is more relevant to our questions than the total variation distance (though we now include a version with this in the supplement).

Minor comments (there are a lot of gaps in the line numbers, so these coordinates are approximate)

- 130: Missing a period "variants To"

Thank you. This has been corrected.

- 156: "In this case, the μ_t are also the same" You assume this in the previous sentence; I think you mean "the m_t "?

Apologies for the confusion. We meant μ_t , not m_t . The point we were trying to make was that the simplification in Equation 5 follows, mathematically, from the fact that Φ^t and μ_t each take on a value that does not change with t when all three factors (population size, selection coefficient, and mutation rate) are constant. However, we agree that the wording was confusing given that we specify verbally in the previous sentence that the mutation rate is constant, so we have removed this phrase from the main text.

- 216: "sites with population frequencies above and below this threshold are represented in the sample" I couldn't figure out what this passage is supposed to mean, please clarify.

What we meant to communicate was that if the sample is large relative to the threshold frequency x_γ^* (i.e. if $2nx_\gamma^* \gg 1$), then sites where the derived allele is found in less than $2nx_\gamma^*$ copies should largely be those for which the population allele frequency is less than x_γ^* , while sites where the derived allele is found in more than $2nx_\gamma^*$ copies should largely be those for which the population allele frequency is greater than x_γ^* . Thus, in sufficiently large samples the shape of the sample site frequency spectrum will reflect the shape of the population SFS.

On the other hand, if the sample is small relative to the threshold frequency (i.e. if $2nx_\gamma^* \lesssim 1$), then most sites with scaled coefficient γ will not make it into the sample at all. Among the few that do, it is most likely that this is because they are among the very lucky few that have population allele frequencies greater than x_γ^* , or else they would not have made it into the sample. In this case, the sample frequency spectrum does not capture the decay seen in the population frequency spectrum.

This is related to the editor's request that we consider the impact of age information on inferences in small samples. We have now updated the text to give a clearer explanation, emphasizing that at this point in the paper we are thinking only about samples that are large in the $2nx_\gamma^* \gg 1$ sense, and pointing readers to later in the paper, where we consider smaller samples.

- 314: "Poisson splitting argument" I think this is usually called Poisson thinning?

Both terms are used in the literature, and we think that Poisson splitting is more appropriate here. Thinning generally refers to the case where events in a Poisson process are randomly removed with some fixed probability, leaving behind a single Poisson process with a reduced rate. Splitting generally refers to the case where events in a single Poisson process are randomly assigned to one of several categories, with potentially different probabilities for the different categories. This generates several 'descendant' Poisson processes whose rates that sum to the rate of the original process. Our situation is the latter, so we think that "splitting" is the appropriate term.

Reviewer #3

This is a very nice and important paper, especially in the current context, and I think it does a good job of exploring confounding factors by also examining a case of exponential growth under a reasonable model. I only have some minor comments on the manuscript.

Thank you for your encouraging words, we appreciate your positive feedback!

1) It took me a while to realize that the denominator of Equation 2 is in fact just $\sum_a m_a$. *It might be worth just pointing out that, of course, the SFS is the sum over all ages that contribute to frequency i .*

Yes, that is correct. We have added a sentence in the main text that now mentions this point: “Notably, $\Phi_{2n}^0[i] = \sum_{a=1}^{T_{\max}} m_a[i]$, i.e., the i^{th} entry in the present-day frequency spectrum is simply a sum over contributions from mutations that arose in all prior generations.”

2) *Also, related to equation 2, is it surprising that the population size at each age does not enter as a “prior” in Equation 2? i.e. it seems like more mutations enter when there is a larger population.*

It does seem surprising at first if one is used to thinking in terms of the evolution of the frequency in the population. In that case, one typically tracks the entire population up until the moment of sampling, so the mutational input scales with the population size. Here, we track only a subsample of the population, so we only care about mutations that arise within that sample.

Another way of putting this is that, yes, there are $2N\mu$ new mutations in the population every generation, but only a fraction $\frac{n}{N}$ of them will arise within some subset of $2n$ chromosomes, so only $2N\mu\frac{n}{N} = 2n\mu$ arise in the sample. The effect of the population size still manifests via the rate of drift, however, because the rate at which mass is moved between adjacent bins of the sample SFS scales inversely with N .

Thus, while the classic “population first” perspective of the diffusion at least seems to suggest that the reason there are more segregating sites in samples taken from larger populations because there are more mutations coming into the population every generation, the “sample first” perspective of the moments machinery suggests that it’s really because segregating sites are lost from the sample more slowly in a larger population. This is more consistent with the coalescent perspective, where the reason there are more segregating sites is because you have to wait longer in between coalescent events (which leads to more mutations that are still segregating in the present).

3) *Although the authors provide some intuition behind Equation 3, I assume they derived it from Equation 1 rather than conjured it, and it would be nice for the authors to detail that derivation in an Appendix.*

We now include Section S3 in the Supplement detailing this derivation.

4) *It might be nice to plot a figure like S6 under the non-constant model, just to get some intuition about what happens under that model and how it changes things.*

Agreed. We now include Figure S10 in the Supplement showing this pattern. As expected, we observe the same pattern as in the constant size case.

Associate Editor

It would be helpful to mention some of the commonly used implementations of PRF based DFE inference in the intro: e.g. DFEalpha (Keightley and Eyre-Walker 2007), polyDFE (Tataru et al. 2017), fastDFE (Sendrowski & Bataillon 2024) in the intro.

This is a great point, we have included these citations with an appropriate sentence in the main text (lines 53-55).

Perhaps I missed this, but it was not clear to me what n was assumed in the simulation test (figs 4 & 5). Was this the $n=100$ as in the analytic comparison in Fig. 3?

More generally, only large samples ($n \sim 100$) are considered. Since SFS based inference of the DFE for non-human data typically involve much smaller samples ($n \sim 10$, e.g. <https://doi.org/10.1534/genetics.116.188102>), it would be helpful to show the sample size dependence of the extra info contained in allele age by adding Results for $n=10$ to Fig 4 & 5.

Yes, you are correct and thank you for clarifying. We have now included the number of samples ($n = 100$) used in creating Figures 4-6 in the captions.

This is a valuable suggestion. We now validate our method in a smaller sample size, $n = 10$. We chose to append text pertaining to the results from this analysis to the section with the MAF thresholding, as it seemed like a natural extension to the message in the paper (lines 334-337). For instance, observing a segregating site in a small sample of size $n = 10$ implicitly imposes a threshold of $x^* = 0.05$, which in turn, indicates that we expect to see information gain with $\gamma = 10$. This is what we observe in Figure 5b.

Fig 3: Although this is given in the figure, I think would help to spell out in the legend that the blue and dashed lines correspond to approximated and true allele distribution. I also wonder whether it would be more useful to express the x axis in $2N$ generations.

Yes, these are useful suggestions and will help with readability. We have now updated the legend to include 'approx.' and 'true' labels for the lines and changed the scale of the x axis to be in units of $2N$ generations.

Please also consider the reviewer's suggestions that "total variation would be a better way than KL divergence since it is interpretable as a percentage"

As outlined in our response to the Reviewer #2, we believe the KL divergence is a more appropriate measure for our argument. We now explain more clearly how to interpret it in this context.

December 12, 2024

RE: GENETICS-2024-307711

Dr. Jeremy J. Berg
The University of Chicago
Department of Human Genetics
5801 S Ellis Ave
Chicago, Illinois

Dear Dr. Berg:

Congratulations! We are delighted to inform you that your manuscript entitled "Allele ages provide limited information about the strength of negative selection" is acceptable for publication in GENETICS. Many thanks for submitting your research to the journal.

To Proceed to Production:

1. Format your article according to GENETICS style, as discussed at <https://academic.oup.com/genetics/pages/general-instructions>, and upload your final files at <https://genetics.msubmit.net>.
2. Your manuscript will be published as-is (unedited-as submitted, reviewed, and accepted) at the GENETICS website as an Advanced Access article and deposited into PubMed shortly after receipt of source files and the completed license to publish. Please notify sourcefiles@thegsajournals.org if you do not wish to publish your article via Advanced Access.
3. We invite you to submit an original color figure related to your paper for consideration as cover art. Please email your submission to the editorial office or upload it with your final files. You can submit a small-sized image for evaluation, and if selected, the final image must be a TIFF file 2513px wide by 3263px high (8.375 by 10.875 inches; resolution of 600ppi). Please avoid graphs and small type.

If you have any questions or encounter any problems while uploading your accepted manuscript files, please email the editorial office at sourcefiles@thegsajournals.org.

Sincerely,

Konrad Lohse
Associate Editor
GENETICS

Approved by:
Nicholas Barton
Senior Editor
GENETICS

note: Please add jnls.author.support@oup.com and genetics.oup@kwglobal.com (or the domains @oup.com and @kwglobal.com) to your email program's "safe senders" list. You will be contacted by both at various points during the production process.